

# Mapping geochemical anomalies by accounting for the uncertainty of mineralization-related elemental associations

Jian Wang[1], Renguang Zuo[2], Qinghai Liu[1]

[1]College of Earth Sciences, Chengdu University of Technology, Chengdu, 610059, China
[2]State Key Laboratory of Geological Processes and Mineral Resources, China University of Geosciences, Wuhan, 430074, China

*Correspondence to*: Renguang Zuo (zrguang@cug.edu.cn)

**Abstract.** Geochemical mapping is a fundamental tool for elucidating the distribution and behaviour of economically significant elements, and providing valuable insights for geological processes. Nevertheless, the quantification of uncertainty
associated with geochemical mapping has recently become a subject of widespread concern. This study presents a procedure, primarily involving the determination of homogeneous clusters, the recognition of elemental associations for each cluster, and the identification of geochemical anomalies, with the aim to account for the uncertainty of elemental association in geochemical mapping. To illustrate and validate the procedure, a case study was conducted wherein stream sediment geochemical samples from the northwestern Sichuan Province, China were processed to map anomalies associated with
disseminated gold mineralization. The results indicate: (1) the representativeness of elemental association for the underlying geological process is an important source of uncertainty for geochemical mapping, (2) the procedure presented here is effective to incorporate the uncertainty of elemental association in geochemical mapping, and (3) the study area can be classified into two clusters, each characterized by unique elemental associations that align well with the distribution of Paleozoic and Triassic lithological units, respectively. Furthermore, the region still holds great potential for the discovery of
gold deposits, particularly in areas proximal to known mineralization sites.

## 1 Introduction

Geochemical mapping plays a vital role in understanding geological processes, discerning the distribution and behaviour of economically significant elements, and facilitating the assessment of the environmental impact of human activities (Bölviken et al., 1990; Cocker, 1999; Pearce et al., 2005; De Vivo et al., 2008; Grunsky et al., 2009; Hou et al., 2015; Wang et al., 2016;
Talebi et al., 2019a; Zuo et al., 2019; Sammon et al., 2022). For example, the mapping of Sr- and Pb-isotopic variations in ocean floor basalts enables the identification of geographically distinct compositional reservoirs within the Earth's mantle (Hart, 1984). In particular, the significance of geochemical maps in mineral exploration, which involves assisting in making informed decisions regarding exploration priorities by identifying concentrations of valuable elements, has been widely recognized (e.g., Rose et al., 1979; Cheng, 2007; Reimann et al., 2007; Carranza, 2008; Xie et al., 2008; Reimann et al.,
30  2016).



Geochemical mapping entails the systematic collection of geochemical samples and processing of geochemical data through multiple steps, with the purpose of mapping spatial variations of geochemical elements and identifying anomalies patterns that may reflect critical geological processes beneath the Earth's surface (Smith and Reimann, 2008; Zuo et al., 2016, 2021;
Grunsky and de Caritat, 2020). Geochemical mapping typically involves four sequential steps: (1) identifying the indicative element or element association that is characteristic of the targeted geological process (e.g., mineralization), (2) predicting or simulating the spatial distribution of the indicator being studied, (3) enhancing and delineating the geochemical signatures of interest, and (4) evaluating the geological significance of the geochemical signatures and their potential to indicate noteworthy geological events (Carranza, 2008; Grunsky and de Caritat, 2020; Wang and Zuo, 2022). It is important to note
that the distinctive geochemical signatures of geological bodies (a.k.a. geochemical anomalies), produced by specific geological processes, can be frequently obscured by subsequent geological or non-geological processes prevailing at the Earth surface (Carranza, 2008; Cheng, 2012; Talebi et al., 2019b; Yousefi et al., 2019). In addition, geological processes that occur across different spatial and temporal scales tend to interact with each other in a multiplicative way. This can result in nonlinearity, heterogeneity, and a mixing of patterns in the resulting geochemical signatures (Cheng, 2012). The scale-
dependent nature and the potential involvement of various heterogeneous geological processes present considerable challenges for geochemical mapping, thereby imposing limitations and uncertainties onto its effectiveness in identifying relevant patterns. Properly addressing the uncertainty is hence crucial to leverage geochemical mapping to understand geological processes and make informed decisions in mineral resource prediction (Wang and Zuo, 2018; Sadeghi, 2021; Zuo et al., 2021a). Previous studies have explored certain aspects of uncertainty that arise from the aforementioned steps involved
in geochemical mapping for mineral deposits discovery (Costa and Koppe, 1999; Wang and Zuo, 2018, 2022; Ersoy and Yunsel, 2019; Chen et al., 2021; Sadeghi, 2021; Liu and Carranza, 2022; Wang et al., 2022; Sadeghi and Cohen, 2023). However, there has been limited research focusing on the uncertainty associated with determining the elemental association as a proxy for the targeted geological process.

Elements tend to be associated due to similar relative mobility in a certain geological process that occurs in unique chemical
and physical conditions, which influence the preferential incorporation or enrichment of certain elements (White, 2020). For example, copper and gold frequently occur together due to their similar geochemical behaviour and affinity for certain geological processes, such as the porphyry copper-gold mineral systems (Sillitoe, 2010). Other notable instances can be found in the elemental association of nickel-cobalt within magmatic sulfide mineral systems, uranium-thorium in sandstone-hosted or vein-type uranium deposits, as well as the gold-silver-arsenic-antimony-mercury association observed in
epithermal gold mineral systems (Pirajno, 2008; Robb, 2020). Note that certain elements may maintain consistent associations across a broad range of geological conditions, whereas others may coexist during most processes in deep-seated environments but become separated in surficial environments (Rose et al., 1979). Grunsky and de Caritat (2020) emphasized that stoichiometry governs the interrelationships among elements in geochemical data, thereby giving rise to distinct structural patterns within the data. Therefore, geological processes can be recognized by a continuum of variable responses.



In this context, a linear model of elements is commonly accepted as a suitable approach to capture the stoichiometry of rock-forming minerals and the subsequent processes (e.g., hydrothermal fluids, weathering) that bring about modifications in mineral structures (Grunsky and de Caritat, 2020; Grunsky et al., 2023). Multivariate statistical methods, such as principal component analysis (PCA), are usually applied to multielement geochemical data to identify the dominant components that generally reflect features related to mineralogy and depict geological processes. For instance, Grunsky and Kjarsgaard (2016)

demonstrated the usefulness of PCA for statistically identifying the distinct geochemical kimberlite phases, which lead to efficiencies in the economic evaluation of kimberlite for diamonds in Saskatchewan, Canada; Mueller and Grunsky (2016) utilized min/max autocorrelation factor analysis on till geochemical survey data collected over the Melville Peninsula, Nunavut, Canada, and effectively predicted the underlying bedrock lithologies and recognized the associated glacial transport processes. Given its remarkable capability to capture nuanced and nonlinear interrelationships among model

variables, machine learning has also been employed to identify significant elemental associations that can serve as representations of underlying geological processes (Zuo, 2018; Grunsky et al., 2023). For example, Wang et al. (2022) utilized a machine learning technique called recursive feature elimination to identify the elemental association patterns that serve as indicators for distinct types of tin mineralization.

During a geochemical survey conducted within a designated area, various geological processes often manifest in distinct

local regions due to the difference of geological conditions, and even within the same area, multiple processes can overlap and intertwine with each other. In a magmatic-hydrothermal gold mineral system (e.g., Masara gold district, Mindanao, Philippines), for example, different types of gold mineralization can take place at different stages and areas, as magmatic fluids evolve and interact with the wall rocks and outer fluids (Robb, 2020). The early high-temperature stage is characterized by porphyry-style mineralization, located at the core of the system directly above the underlying magma

chamber, which primarily yields disseminated gold-copper sulfides such as chalcopyrite, bornite and molybdenite; in the intermediate stage, epithermal quartz-adularia-gold vein mineralization is prominent, forming a ring-shaped zone surrounding the porphyry core, which produced native gold and sulfides like pyrite, galena and sphalerite; the late stage is typically associated with low-sulfidation epithermal mineralization, occurring further outward from the core, which is characterized by quartz-carbonate veins with high Au/Ag ratios, and gold occurring as electrum with minerals like pyrite,

marcasite, stibnite and realgar (Pirajno, 2008). Hot spring gold mineralization can also occur when the remaining magmatic fluids mix with meteoric water at the surface and cool further. It is important to note that the heterogeneous zonation observed in the mineral system can be disrupted by structural controls such as faults, which serve as pathways for mineralizing fluids. In such a complicated context, relying solely on a single group of elements as a proxy for underlying geological processes can inevitably lead to uncertainties in the resulting geochemical patterns. Consequently, how to address

the uncertainty arising from the representativeness of elemental associations in geological processes becomes a significant concern when utilizing geochemical mapping to comprehend the processes and aid in mineral resource prediction.





This paper proposes a workflow that combines fuzzy clustering, PCA and geochemical anomaly identification methods to reduce the uncertainty of elemental association in geochemical mapping. To illustrate and validate the procedure, a case

study was conducted wherein stream sediment geochemical samples from Northwest Sichuan Province, China were processed, with the aim of delineating anomalies associated with sediment-hosted disseminated gold mineralization.

## 2 Methods

### 2.1 The general workflow

The general workflow that accounts for the uncertainty of elemental association in geochemical anomalies mapping (Fig. 1)

primarily consists of four parts:

(1) identifying homogeneous regions through fuzzy clustering;

Prior to cluster analysis, individual elements in a selected multivariate geochemical survey dataset are spatially interpolated. The interpolated maps are subsequently utilized as input for the fuzzy clustering method, commonly fuzzy $c$-means, to obtain membership values maps. The number of membership values maps is equal to that of clusters determined by

optimization metrics, such as the Silhouette index (Rousseeuw, 1987), gap statistics (Tibshirani et al., 2001) or cluster validity index (Xie and Beni, 1991). The homogeneous local regions can then be determined by the largest membership value for each grid cell.

(2) determining elemental associations for each region, which serve as representative indicators of the underlying targeted process;

The geochemical survey data is initially partitioned into distinct subsets based on the criteria of sample assignment to specific clusters. Subsequently, each subset of data undergoes PCA, enabling the examination of elemental associations through a biplot analysis. By identifying a distinctive set of elements for each subset, representative of the geological processes of interest, a comprehensive understanding of targeted geological phenomena with uncertain elemental associations can be achieved.

(3) recognizing multivariate anomalies based on each elemental association;

A geochemical anomalies identification algorithm (e.g., local singularity analysis by Cheng (2007), deep autoencoder network by Xiong and Zuo (2016)) was firstly applied to the interpolated map of each element to obtain univariate anomaly patterns. Multivariate anomaly patterns were derived by integrating relevant univariate anomalies through PCA. For each potential elemental association linked to the underlying geological process, PCA was applied solely to the subset of elements

within that assemblage. The first principal component score map, which captures the highest amount of variation, was retained to represent the multivariate anomaly patterns (e.g., Cheng, 2007). Note that the number of multivariate anomalies maps is the same as the number of clusters.

(4) integrating alternative anomaly patterns to generate a comprehensive map of anomalies.





The multivariate anomalies maps can be further integrated into a comprehensive anomaly map using a linear weighting
model. The weights assigned to each map correspond to the membership values obtained from fuzzy clustering, specifically
to the cluster from which the map was derived. Since fuzzy clustering inherently normalizes memberships, these values
intrinsically account for each domain's spatial representation and influence.

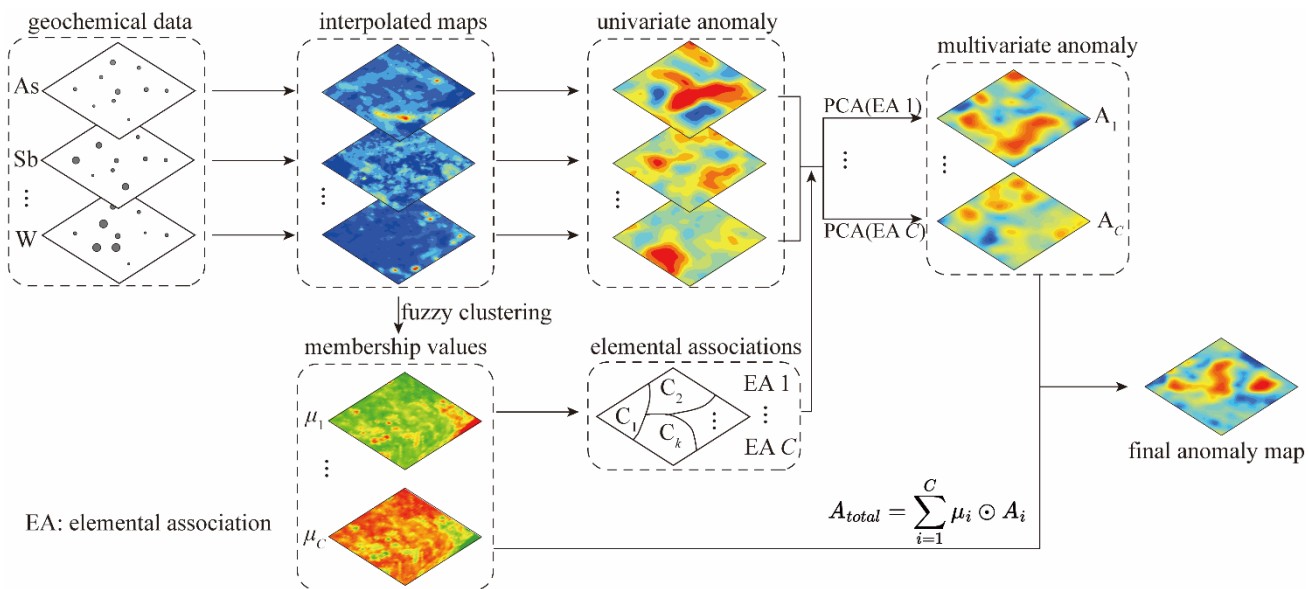

**Figure 1: The general workflow for geochemical anomalies mapping by accounting for the uncertainty of elemental association.**

### 2.2 Fuzzy $c$-means clustering

The aim of clustering is to divide a set of $N$ data points into $C$ clusters, such that data points within a cluster exhibit
similarity while in different clusters are dissimilar. Clustering serves the purpose of extracting a set of cluster prototypes,
enabling a compact representation of the dataset with several homogeneous subsets (Kaufman and Rousseeuw, 2009). Fuzzy
set theory assumes that data points may not belong exclusively to a single set, but rather have a degree of membership
uncertainty that can be addressed through the use of a membership function (Zadeh, 1965). Integration of fuzzy logic with
data mining techniques has emerged as a fundamental aspect of soft modeling to address uncertainty (Bezdek, 2013). Fuzzy
$c$-means (FCM), firstly developed by Dunn (1973), is such an unsupervised soft clustering technique that allows data points
to be classified into multiple clusters with varying degrees of membership (Bezdek et al., 1984). FCM is an iterative
algorithm that computes cluster centers and membership values to minimize the following objective function

$$\mathcal{L}_m = \sum_{i=1}^{C} \sum_{j=1}^{N} \mu_{ij}^{m} D_{ij}^2, \tag{1}$$



where $C$ denotes the number of clusters, $N$ is number of data points, $m(m > 1)$ is a hyper- parameter that controls the degree of fuzzy overlap, which refers to how fuzzy the boundaries between clusters are; $\mu_{ij}$ is a continuous value between 0 and 1, and represents the degree of membership of the $j$th data point in the $i$th cluster; $D_{ij}$ is the distance between the $j$th data point $x_j$ and the center of the $i$th cluster $c_i$, for which the Euclidean distance is commonly used such that $D_{ij} = \left\| x_j - c_i \right\|^2$. Note that for a given data point, the sum of its membership values for all clusters is constant one, namely

$$\sum_{i=1}^{C} \mu_{ij} = 1, j = 1,2,\cdots, N, \tag{2}$$

The classical FCM computes distances $D_{ij}$ between data points and cluster centers using a Euclidean distance metric. However, other dissimilarity metrics can also be employed to establish alternative clustering algorithms. For instance, Gustafson and William (1978) presented a fuzzy clustering algorithm that computes distances using a Mahalanobis distance metric, which enables to account for correlations and variations in multiple dimensions or variables. The implementation of FCM closely resembles that of $k$-means, and for specific algorithmic details, one can refer to the work of Suganya and Shanthi (2012).

A key advantage of FCM lies in its flexibility in assigning gradual memberships to account for uncertainty. Hence, FCM has been one of the most widely used fuzzy clustering algorithms in data science and machine learning applications (e.g., Fatehi and Asadi, 2017; Benjumea et al., 2021; Zhang et al., 2021).

### 2.3. Derivation of the comprehensive anomaly map

Assuming the $C$ elemental associations are $EA_i(i = 1,2,\cdots, C)$, and the multivariate anomaly map for elemental association $EA_i$ is $A_i$, the comprehensive anomaly map can be derived through the following formula

$$A_{total} = \sum_{i=1}^{C} \mu_i \odot A_i, \tag{3}$$

where $\mu_i$ represents the membership values map for the $i$th cluster, which serves as an indicator of the confidence level associated with the anomaly map $A_i$ portraying the underlying targeted process; the operator $\odot$ denotes 'Hadamard product', that is, the element-wise product. It is a binary operation that takes in two matrices of the same dimensions and returns a matrix of the multiplied corresponding elements.

## 3. Study area and data

### 3.1. Geological setting

The study area is situated in the northwestern region of Sichuan Province, China, encompassing a longitude range of 103°4′E to 104°36′30″E and a latitude range of 103°4′E to 104°36′30″E (Fig. 2).



Located at the intersection of the Yangtze plate, North China plate and Songpan-Ganzi terrane, this area has been distinguished by active tectonic and magmatic processes throughout geological histories. These long-lived crustal dynamics exert significant controls on the formation and widespread distribution of gold mineralization observed across the region. Previous studies have revealed a strong correlation between the emplacement of large gold deposits and the presence of NW-SE extending major tectonic faults, as well as the intersection of multiple faults and ring-shaped fault systems (e.g., Zhao,

1995; Li, 1996; Wang et al., 2003; Liu et al., 2010). In this area, one can find stratigraphic units ranging from the Proterozoic to the Cenozoic. The distribution of these units is evidently controlled by regional faults, and they are also prone to undergoing metamorphism. The Triassic strata, which covers approximately 73% of the study area, predominates in the western and northern parts. It mainly consists of metamorphosed sandstones and slates that are interbedded with occasional volcanic rocks and limestone. These strata primarily represent shallow sea slope turbidite sedimentary environment and play

a significant role as the main sources of materials for gold mineralization, as demonstrated by the isotopic and rare earth element geochemistry (Zheng et al., 1990; Chen, 1998; Wang et al., 2004; Zhang, 2014). The igneous rocks, which appear infrequently at the surface and are primarily confined to the southeastern portion of the study area, consist of various types such as granites, granodiorites, and monzogranites. Previous studies have indicated that hypabyssal calc-alkaline igneous rocks from the late Indonesian to Yanshanian period play a crucial role in the generation of hydrothermal fluids and the

creation of geodynamic conditions that facilitate the remobilization and concentration of gold in this region (e.g., Li, 1996; Liu et al., 2010).

The predominant type of gold deposits discovered in this area is sediment-hosted disseminated gold deposits, exemplified by the Dongbeizhai and Manaoke gold deposits. These deposits are primarily found within the Triassic marine sequences. They are characterized by the presence of microscopic and/or dissolved gold, as well as a mineral association for epithermal

mineralization that includes arsenopyrite, pyrite, stibnite, among others. Furthermore, studies imply that there are variations in the geological and geochemical characteristics of different gold deposits due to individual differences in tectonic settings and geological conditions. Consequently, these heterogeneities pose challenges for the processing of geochemical data (Li, 1996; Chen, 1998; Chen et al., 2004; Deng et al., 2023).

### 3.2. Geochemical survey data

The geochemical data utilized in this research is derived from the China's National Geochemical Mapping Project, which was initiated in 1979 and has played a critical role in mineral exploration in China (Xie et al., 1997). It comprises 3461 composite stream sediment samples collected at a density of one sample per 4 km². Each sample was analyzed for 39 major, minor, and trace elements/oxides, that is, Ag, As, Au, B, Ba, Be, Bi, Cd, Co, Cr, Cu, F, Hg, La, Li, Mn, Mo, Nb, Ni, P, Pb, Sb, Sn, Sr, Th, Ti, U, V, W, Y, Zn, Zr, SiO2, Al2O3, Fe2O3, K2O, Na2O, CaO and MgO. For comprehensive information

regarding sample preparation, analytical methodologies, detection limits, and quality control, please refer to the works of Xie et al. (1997) and Wang et al. (2011).



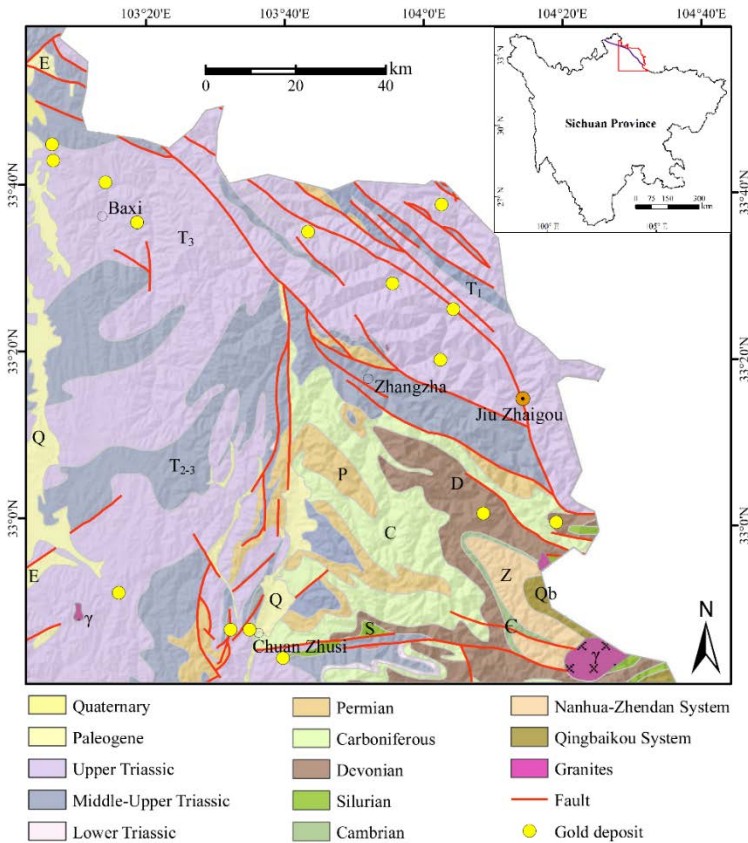

**Figure 2: Simplified geological map of the study area in northwest Sichuan Province, China (after Wang and Zuo (2022)).**


## 4. Results and discussions

### 4.1. The uncertainty of elemental associations related to gold mineralization

Zuo et al. (2021b) explored the lower-order statistics of the ore-forming element Au in this dataset by using exploratory statistical graphs, including boxplot, histogram, and quantile-quantile plot. The result suggested that the original
concentrations of Au exhibit an evidently positively-skewed and heavy-tailed distribution, implying that the geochemical data might originate from more than one geological process, with gold mineralization imposing an important influence in shaping the distribution. A global elemental association, consisting of Au, As, Sb, and Cu, has also been identified in this area by applying PCA for compositional data onto fifteen trace elements (i.e., Ag, As, Au, Cd, Ba, Bi, Cu, Hg, Mn, Mo, Pb, Sb, Sn, W, and Zn). Furthermore, the spatial patterns of this elemental association ascertain its correlation with gold
mineralization and its relationship with the distribution of fault systems that controlled the mineralization (Zuo et al., 2021b). However, relying solely on a single elemental association might not adequately represent the potential mineralization in this




area. This limitation arises from the inherent heterogeneity and multi-stage nature of gold mineralization, as indicated by previous geological studies (e.g., Chen, 1998; Chen et al., 2004; Deng et al., 2023). According to Li (1996), this study area exhibits at least two distinct types of gold mineralization. The first type is predominantly controlled by structures, and is

typically characterized by hydrothermal minerals such as arsenopyrite, stibnite, realgar, orpiment, and microscopic natural gold. The common elemental association observed in this type is Au-As-Sb-Hg. The second type of gold mineralization is primarily controlled by igneous veins. The typical hydrothermal minerals associated with this type include pyrite, arsenopyrite, stibnite, barite, and microscopic natural gold, which are also accompanied by contact metasomatism-derived chalcopyrite and galena, among others. This type of mineralization exhibits an elemental association of Au-As-Sb-Ba-Cu-Pb.

Other studies, such as Chen (1998), Zhao (1999), Wang et al. (2004) and Deng et al. (2023), also suggest that high-temperature hydrothermal fluids play a crucial role in remobilizing and concentrating ore-forming elements. Therefore, elements such as W and Sn can also serve as indicators for gold mineralization in this area. Consequently, while the elemental association Au-As-Sb is commonly observed across the gold deposits in this region, individual deposits exhibit enrichment in certain pathfinder elements that are characteristic of the local mineralization.


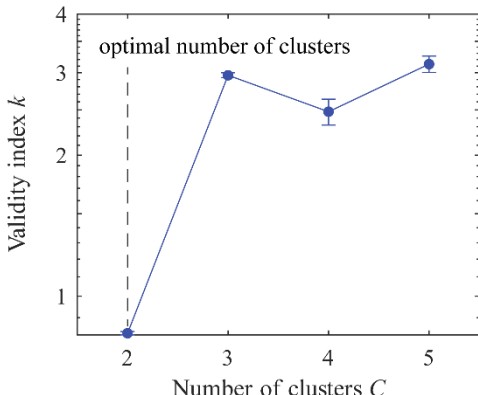

**Figure 3: Optimal cluster number determined by the cluster validity index. Note that a total of 100 experiments were conducted to achieve a robust result.**

Based on the procedure outlined in Section 2.1, we initially applied inverse distance weighting (IDW) to the same fifteen trace elements investigated by Zuo et al. (2021b) (Fig. 3). The cell size was set to 1 km, and the local interpolation utilized a default value of 12 neighbors in ArcGIS. Subsequently, FCM was performed onto the interpolated maps. Various cluster numbers were explored, and the optimal value of 2 was determined based on the cluster invalidity index (Fig. 3). The FCM analysis assigned each grid cell a membership value indicating its degree of belongingness to each of the two clusters (Figs.

4a and b). A cluster label could be specified to each cell with the largest membership value. The clustering results reveal that Cluster 2 is primarily distributed in the southeast of the study area, while Cluster 1 is distributed pervasively throughout the rest of the study area. Cluster 1 mainly reflects the distribution of Triassic sequences, while Cluster 2 mainly reflects the



distribution of Paleozoic sequences (Fig. 2). The marginal plot in Figure 4c indicates that Cluster 2 is characterized by evidently higher concentrations of Au and As. This observation is consistent with the geological knowledge that the

Paleozoic carbonaceous silty shale formation exhibits a high geochemical background in elements related to gold mineralization, and serves as one of the most important sources of materials for gold mineralization in this area (e.g., Li, 1996; Zhang, 2014).

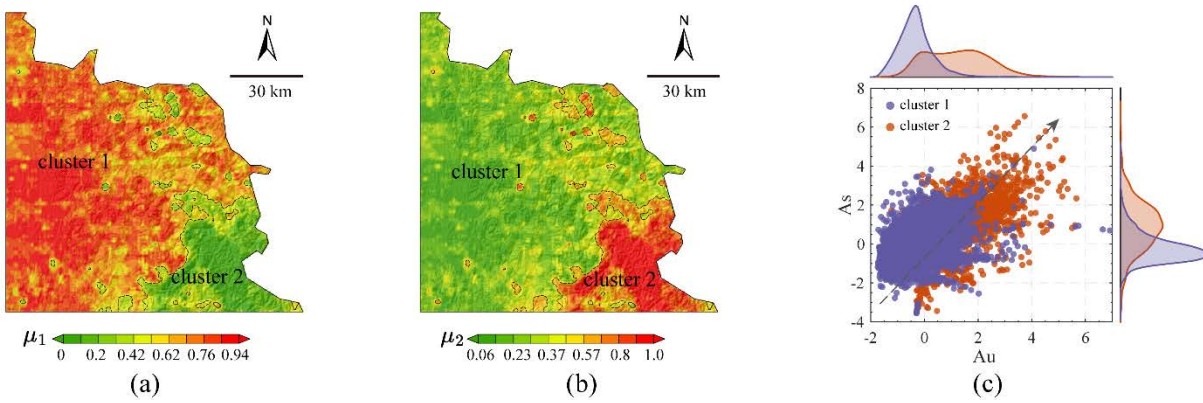

**Figure 4: Fuzzy *c*-means clustering of the interpolated maps. (a) Distribution of the fuzzy membership values for cluster 1, (b) distribution of the fuzzy membership values for cluster 2, and (c) marginal plot showing the distribution of data points for Au concentrations against As concentrations. Note that the concentration values were logarithmically transformed and standardized for improved visualization.**

To identify potential elemental associations that indicate gold mineralization in this area, we performed PCA separately on the data from each of the two clusters. The resulting biplot, which depicted the first two principal components, was utilized for visual exploration of the elemental associations. The biplot analysis indicates that the first two principal components account for a total of 55% of the variation within the elements in Cluster 1, while in Cluster 2, the explained variation is 61% (Figs. 5a and b). In a biplot, the angle between two vectors that represent geochemical elements can provide an

approximation of their correlation (Gabriel, 1971). By applying this principle, we can identify the potential elemental association indicative of gold mineralization by examining the relationship between each element and the ore-forming element Au. In addition, we also incorporated geological knowledge regarding expected elemental associations and the distribution of known gold deposits depicted in the biplot to determine an elemental association that closely aligns with the known deposits. For Cluster 1, the elemental association identified is Au-W-As-Sb-Ba-Hg, while for Cluster 2, it is Au-As-

W-Sn-Sb-Hg-Pb-Bi (Figs. 5a and b). These elemental associations demonstrate strong consistency with the aforementioned geological knowledge and can effectively predict the majority of known gold deposits.





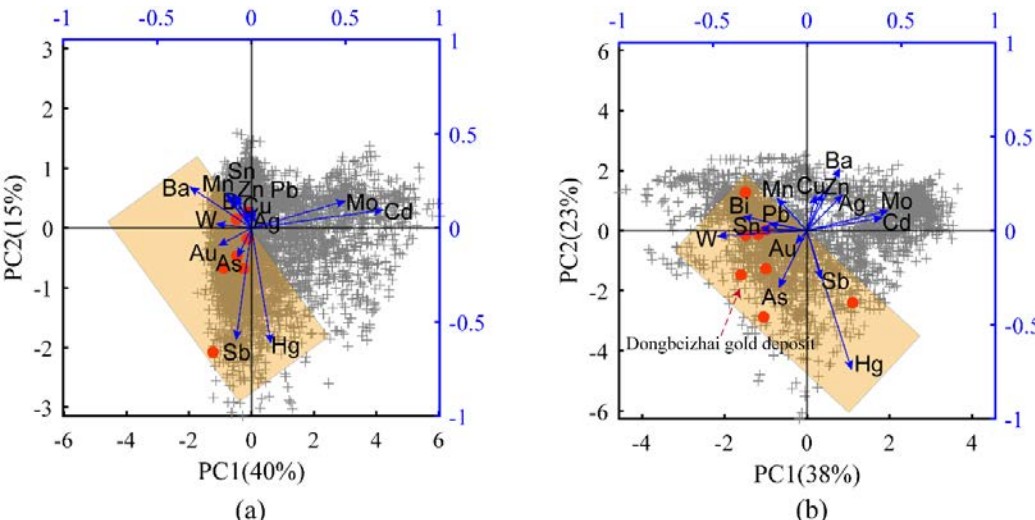

**Figure 5: Biplots of the first two principal components obtained from PCA of (a) Cluster 1 and (b) Cluster 2. Note that the red points represent the projections of the known gold deposits. The shaded orange rectangle encompassed the elements that show good correlations with the ore-forming element Au.**

### 4.2. Mapping single- and multi-element anomalies patterns

The interpolated maps of the elements that show correlations with ore-forming element Au for both clusters (i.e., Au, As, W, Sb, Sn, Hg, Ba, Pb, Bi) were used further for mapping local singularities. The effectiveness of local singularity exponents in enhancing anomaly patterns by mitigating the mask effect of heterogeneous local backgrounds has been well established (e.g., Cheng, 2007; Chen and Cheng, 2016; Li et al., 2017; Gonçalves et al., 2018; Wang et al., 2018; Xiao et al., 2018). For detailed theoretical and algorithmic information on local singularity analysis (LSA), please refer to Cheng (2007). Prior to calculating the singularity exponent, several model parameters concerning the sliding window need to be specified. In this study, we utilized a series of square windows, with varying half window sizes ranging from 1 km to 13 km at 2 km intervals. Figure 6 illustrates the distributions of singularity exponents estimated using the sliding window-based technique. It can be observed that the local patterns indicated by singularity exponents are clear and remain scarcely unaffected by the heterogeneous geological background. The singularities for the primary indicative elements, such as Au, As, W, Sb, exhibit strong spatial correlations with the distribution of known gold deposits. In addition, the distribution of singularity for elements Au, As, Sb and Bi, which are often associated with hydrothermal systems and can exhibit significant mobility and volatility, can clearly exhibit the distribution of geological structures in this area. It is noteworthy that all the elements studied here exhibit distinct anomaly patterns (i.e., positive singularity), regardless of their strength, in the vicinity of the giant Dongbeizhai gold deposit (highlighted by white solid rectangles in Fig. 6). In contrast, certain gold deposits may not display anomaly patterns in the singularity maps generated for specific elements. However, in the maps of other elements, discernible anomaly patterns can be identified for these gold deposits, as indicated by the white dashed rectangles in Figure 6.



Such observation highlights the inherent uncertainty associated with indicative elements in relation to localized gold mineralization.



**Figure 6: The distribution of local singularity exponents for single element within the set of elements that show correlation with Au**
**for both Cluster 1 and Cluster 2: (a) Au, (b) As, (c) W, (d) Sb, (e) Sn, (f) Hg, (g) Ba, (h) Pb, and (i) Bi.**



To delineate the comprehensive anomaly patterns with the combined elements for both Cluster 1 and Cluster 2, we applied PCA onto the singularity exponents for elements in the identified elemental associations (Fig. 5). The first principal components account for 46% and 53% of the total variance for Cluster 1 and Cluster 2, respectively. The multi-element

anomaly patterns for the two clusters exhibit similarities in general, in that they align well with the geological structures and can effectively predict known gold mineralization. However, there are variations in the local details of the anomaly patterns across different clusters. Furthermore, the overall anomaly intensity for Cluster 1 (Fig. 7a) is slightly higher compared to Cluster 2 (Fig. 7b). Notably, the resulting map for Cluster 1 exhibits distinct multi-level patterns in certain local areas, which are very weak or even absent for Cluster 2. The multi-element anomaly for the two clusters were further integrated into a

comprehensive anomaly map using a linear weighting scheme that utilized the fuzzy membership values as weights. The resulting map preserves the common patterns that display good correlation with known gold mineralization (Fig. 7c). More importantly, the integrated singularity map also underscores the importance of detecting underlying geological structures and mineralization patterns in the western portion of the study area.

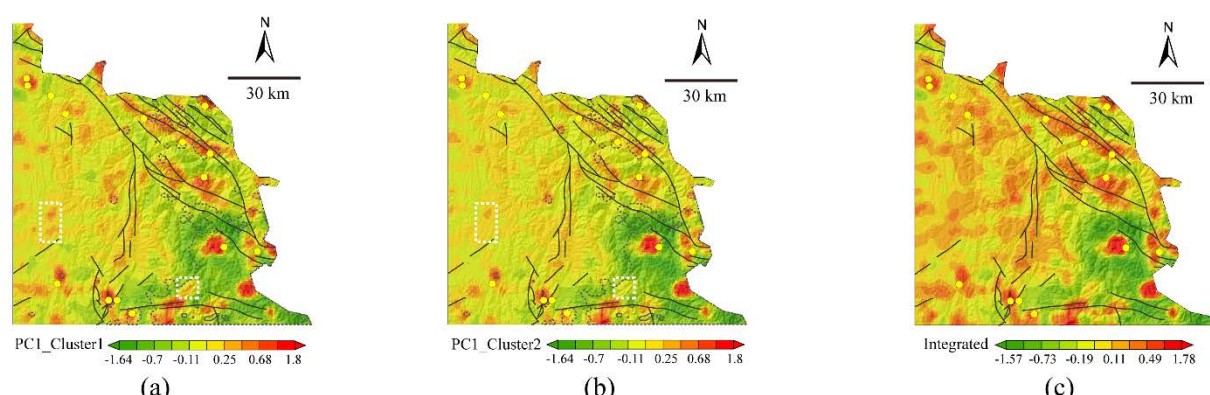

**Figure 7: The first principal component scores showing the distribution of combined singularities obtained by applying PCA to singularities of (a) Au-W-As-Sb-Ba-Hg for Cluster 1, and (b) Au-As-W-Sn-Sb-Hg-Pb-Bi for Cluster 2. The comprehensive map of anomalies patterns integrated from the combined singularities maps based on fuzzy membership values is shown in (c). Note that local patterns enclosed by white dashed rectangles in (a) and (b) indicate the difference between the multi-element anomaly map**

**for Cluster 1 and 2.**

## 4.3. Model evaluation

To evaluate the performance of the result presented in Figure 7c, we additionally identified multi-element anomaly patterns while disregarding the uncertainty of elemental associations. This was then used as the benchmark for performance

comparison, referred to hereafter as the "global reference case". The elemental association adopted for the global reference case is Au-As-Sb-Cu, with the purpose of being aligned with the study of Zuo et al. (2021b). The multi-element anomaly patterns obtained for the global reference case (Fig. 8a) exhibit strong spatial correlation with the geological structures and





known gold mineral deposits. However, differences can also be easily observed when comparing it to the results obtained from the procedure that takes into account the uncertainty of elemental associations, which will be referred to as the "case

with uncertainty". For example, there is no anomaly present in the vicinity of the easternmost known gold mineralization, as indicated by the white dashed rectangle in Figure 8a. In contrast, clear anomaly patterns associated with this deposit can be observed in the resulting map for the case that considers uncertainty (Fig. 7c). In the present study, the multi-element anomaly patterns for the two cases were verified using the success-rate curves in terms of their capability in predicting the known mineralization, which was obtained by plotting cumulative percentage of known gold deposits against cumulative

percentage of anomaly patterns area (e.g., Carranza, 2008). In general, the result from the case with uncertainty outperforms the global reference case (Fig. 8b). The success-rate curves also suggest that approximately the top 6% of the total area can predict around 54% of known mineralization (point A), regardless of whether we consider the global reference case or that accounts for the uncertainty of elemental association. However, when a larger area is delineated to predict the gold mineralization from 54% (point A) to 83% (point B), the success-rate curve exhibits distinct behaviors between the two

scenarios. The case that considers uncertainty can consistently predict the same proportion of known gold mineralization with a relatively lower percentage of study area than the global reference case. By examining the local areas delineated by the cut-off values corresponding to points A and B for the two scenarios (Fig. 8c), we were able to visually discern the disparities in the spatial distribution of geochemical patterns that contribute to the different performance observed between these two points. The global reference case placed a greater emphasis on the southeastern region, which is characterized by

high geochemical backgrounds for most indicative elements due to the prevalence of Paleozoic gold-enrichment lithologic units. This observation suggests that the procedure proposed in this study might have the potential to mitigate the impact of heterogeneous geochemical backgrounds in geochemical anomaly mapping.

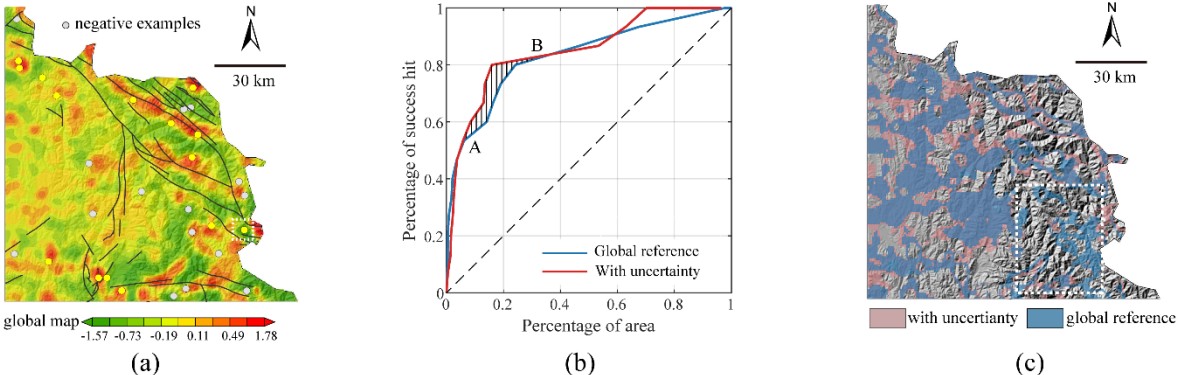

**Figure 8: A comparison made between the performance of the mapped results, predicting known gold mineralization, in two scenarios: the global reference case that does not consider the uncertainty (a) and the case presented in this study that accounts for the uncertainty of elemental association (Fig. 7c). The success-rate curves for these two scenarios are shown in (b), and the areas delineated by the cut-off values corresponding to the two points A and B in (b) for the two scenarios are displayed in (c).**





To derive a quantitative metric for accuracy assessment, this study also utilized the receiver operating characteristic (ROC) curve and the area under curve (AUC) methodology, as described by Fawcett (2006). When constructing a ROC curve, negative examples that represents the absence or non-occurrence of mineralization event are required to be used along with the positive examples (i.e., known gold mineralization) to evaluate the performance of a binary classification model. In addition, studies also suggested that the number of negative examples should be similar to that of positive examples to

ensure a balanced evaluation. We randomly generated a set of negative examples under the constraint that they are located outside the 3 km local neighborhood of known deposits (Fig. 8a). Moreover, the number of negative examples was set to match that of known mineral deposits. The ROC curves in Figure 9a depict the true positive rate (TPR) and false positive rate (FPR) at various classification thresholds for both the case considering the uncertainty of elemental association and the global reference case. The AUC values were determined to be 0.8 for the global reference case, and 0.85 for the case with

uncertainty. Therefore, when compared to the global reference case, the case with uncertainty demonstrates superior overall performance in terms of accurately identifying known gold mineralization while minimizing false positives. Considering the potential uncertainties involved in calculating the AUC value, we proceeded to randomly generate multiple sets of negative examples. Specifically, a total of 300 sets were created to mitigate potential biases or peculiarities that may exist in a single negative example set. The results (Fig. 9b) suggest that the case considering the uncertainty of elemental association can, on

average, outperform the global reference case in predicting known mineralization. Also, it should be noted that the case with uncertainty exhibits higher sensitivity to the selection of negative examples. Given that the elemental association for the global reference case involves a total of four elements, we additionally investigated a scenario where only the top four relevant elements were retained for the two clusters. It is evident from the results (Fig. 9b) that this particular case exhibits superior performance compared to the previous two cases on average. This observation indicates that the incorporation of

certain elements that exhibit weak correlation with the ore-forming element Au may offer limited or even detrimental contributions to the accurate mapping of geochemical anomalies.

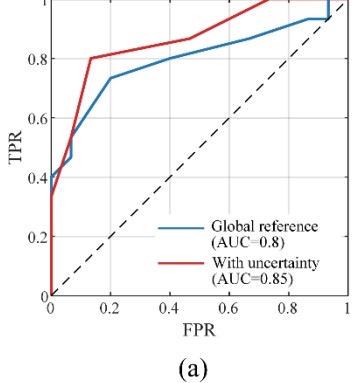

(a)

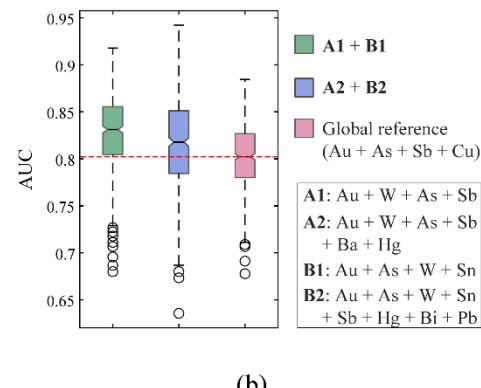

(b)



**Figure 9: (a) The ROC curves and AUC values for the global reference case and the case considering the uncertainty of elemental association, (b) the boxplot showing the AUCs from 300 experiments that sample different negative examples for the two scenarios. Here different combinations of elemental associations for the two clusters were also examined.**

## 4.4. Delineation of significant geochemical anomalies

To further delineate significant geochemical anomalies for guiding subsequent mineral exploration, the weights of evidence method was used to derive the statistical $t$-values, which allows for defining significant anomalies (Bonham-Carter, 1994). The $t$-value serves as a measure of the significance of spatial correlation between point features and polygons, with higher $t$-values indicating stronger spatial correlation. Typically, a $t$-value = 1.96 can be taken to be a threshold above which the spatial correlation can be regarded statistically significant.

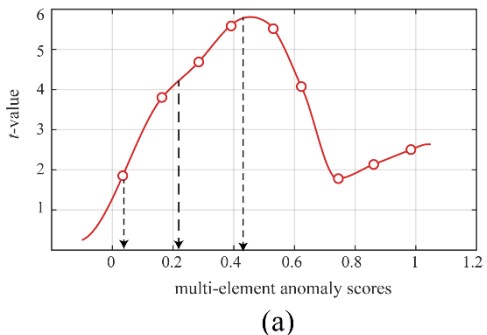

(a)

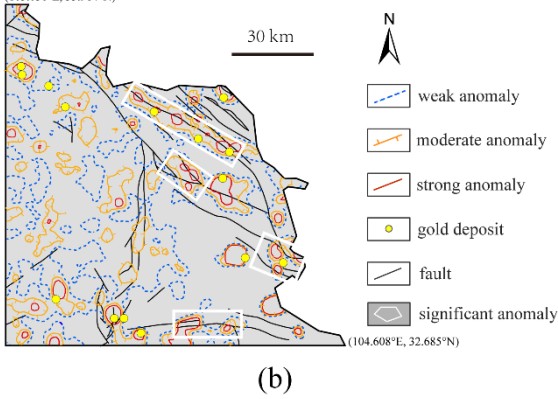

(b)

**Figure 10: The student's t values (a) and the delineated geochemical anomalies (b) based on the multi-element anomaly scores derived from the case considering the uncertainty of elemental association (Fig. 7c).**

The $t$-values for the resulting anomaly scores map (Fig. 7c), as depicted in Figure 10a, demonstrate an increasing trend as the threshold rises from 0 to 0.42, eventually reaching the maximum. It is important to note that the portion of the study area



with an anomaly score ≥ 0.42 occupies only 7% of the total area, yet it contains 60% of the total number of mineral deposits. In addition, the *t*-value reaches 1.96 at an anomaly score of 0.03. The two values 0.03 and 0.42, along with an arbitrarily determined anomaly score of 0.21 nearly at their midpoint, were utilized as thresholds to define the weak anomaly (0.03-0.21), moderate anomaly (0.21-0.42) and strong anomaly (≥ 0.42) (Fig. 10b). The result shows that the delineated patterns are directly associated with the known gold deposits. Notably, most of the known deposits are spatially linked to multi-level anomaly patterns. We also preliminarily delimited some significant anomalies based on the following criteria: (a) presence of multi-level anomaly patterns, (b) proximity to known deposits, (c) proximity to geological structures. The delimited anomalies (Fig. 10b) should be further investigated for undiscovered gold deposits.

## 4.5. The implications and limitations of the procedure for mapping geochemical anomalies under uncertainty

Geochemical patterns arise from dynamic geological systems that are open, nonlinear, complex, and subject to spatial and temporal variations. The intrinsic heterogeneity of these patterns poses challenges in identifying and understanding the underlying component geological processes based on geochemical data, thus leading to inherent uncertainties. Specific geological processes are commonly considered to have the potential to be reliably represented by certain elemental associations. Therefore, in order to address the uncertainties and improve our comprehension of geological processes and mineral resource prediction, it is crucial to identify and analyze the diverse elemental associations present in a given study area. The procedure presented here identified two distinct clusters within the study area, and they are characterized by different elemental associations related to gold mineralization. Cluster 1 covers a significantly larger area and predominantly encompasses the Triassic formations, whereas the other cluster is primarily composed of the Paleozoic lithologic units. These two clusters can be approximately differentiated by the regional Heye fault belt, trending NW-SE, and the Minjiang fault belt, trending S-N, which serve as the boundaries between the above geological units. Studies have also demonstrated the significant influence of regional fault belts in constraining and delineating areas where various geological processes have occurred throughout geological history. These processes encompass sedimentation, magmatic activities, metamorphism, and mineralization events (Wang and Liang, 2004). The presence of regional structures in the area highlights the evident spatial heterogeneity in the geological composition across various regions and throughout different geological time periods. The study area was recognized as a passive continental margin during the late Proterozoic to Paleozoic era, characterized by the development of sedimentary cover layers primarily consisting of terrigenous clastics with minor occurrences of carbonate and siliceous rocks. However, during the Mesozoic era, the area experienced tectonic movements associated with the ancient Tethys, resulting in extensive folding within regions where Paleozoic sequences are distributed. In other areas, intense faulting occurred, accompanied by the deposition of extensive thick flysch sequences during the Triassic period. These flysch sequences have proved to be crucial sources of ore-forming materials (Wang et al., 2003). The regional geochemical analysis suggests that the Paleozoic lithological units are characterized by a higher geochemical background level of Au compared to the Triassic formations (Zhao, 1995). In addition, according to previous studies (e.g., Zhao, 1995), there is a



discernible pattern where the temperature of the mineralization-related fluids increases from north to south. This geological

knowledge can be further supported by the elemental association observed in Cluster 2 in this study, which includes high-temperature hydrothermal elements such as Sn, W, and Bi.

Note that the current procedure only account for the dissimilarity of elemental concentrations during fuzzy clustering, while disregarding the tectonic setting and geological conditions of the data points. Consequently, it is evident that Cluster 2 includes irregular and disconnected areas in addition to the major southeast area that exhibits a high geochemical

background (Fig. 4), although the membership values of these scattered areas are relatively lower. Note that these small areas are characterized by high concentration values for the selected geochemical elements. However, the geological sequence in these areas is Triassic, which differs from that of the southeast area. Therefore, future studies should focus on extending classical fuzzy clustering algorithms to account for geological constraints, or take spatial connectivity into consideration as an additional constraint. We also observe that relying solely on the biplots to determine elemental

associations can introduce additional uncertainty. This is because only part of the variation is explained by the biplot itself, and there is a lack of widely accepted criteria to determine the optimal subset of elements that exhibit a strong correlation with the ore-forming element of interest. Nevertheless, the case study presented here indicates that the procedure that considers the uncertainty of elemental associations provide a promising approach to achieve superior performance in geochemical anomalies mapping compared to the global case where such uncertainty is not taken into account.

**5. Conclusions**

In this study, we have developed a procedure that accounts for the uncertainty of elemental associations as an indicator of the underlying geological process of interest, aiming to improve geochemical mapping. A case study of processing stream sediment geochemical samples to map geochemical anomalies linked to disseminated gold mineralization in the northwestern Sichuan Province, China was presented to illustrate and validate the procedure. Three main conclusions could

be drawn thereby:

(1) determination of an elemental association as an indicator of the underlying geological process is an important source of uncertainty for geochemical mapping;

(2) the procedure outlined in this study, which mainly comprises fuzzy clustering, principal component analysis, and geochemical anomaly identification algorithms, provides an effective framework for addressing the uncertainty associated

with elemental associations in geochemical mapping. Also, note that the procedure allows for the incorporation of alternative methods for fuzzy clustering, determination of elemental associations, and identification of geochemical anomalies, rather than being limited to the methods employed in this particular study. This provides greater flexibility and adaptability to suit different research contexts;

(3) two distinct clusters can be identified within the study area, aligning closely with the distribution of lithological units

impacted by diverse geological processes. Moreover, the procedure presented here demonstrates, on average, superior

performance compared to the global reference case in accurately predicting gold mineralization. The delineated anomaly patterns show potential for the discovery of more gold deposit in this region. It is worth noting that attention should also be paid towards the western areas, where minimal gold deposits have been uncovered thus far. However, weak anomalies persist in these regions, possibly indicative of deeply buried mineralization and underlying structures.


*Code and data availability*. The core code can be obtained by emailing the first author: Jian Wang (jwang@cdut.edu.cn). The authors do not have permission to share data.

*Author contributions*. JW contributed to conceptualization, data curation, formal analysis, funding acquisition, investigation, 470 methodology, validation, visualization, and writing the original draft. RZ contributed to the conceptualization of the study, and reviewed and supervised the work. QL made visualization and writing the original draft. All the authors have read and approved the final manuscript.

*Competing interests*. The authors declare that they have no known competing financial interests or personal relationships that 475 could have appeared to influence the work reported in this paper.

*Disclaimer*. Publisher's note: Copernicus Publications remains neutral with regard to jurisdictional claims made in the text, published maps, institutional affiliations, or any other geographical rep-representation in this paper. While Copernicus Publications makes every effort to include appropriate place names, the final responsibility lies with the authors.


*Acknowledgements*. We thank Xueqiu Wang at Institute of Geophysical and Geochemical Exploration, China to provide the geochemical data.

*Financial support*. This research benefited from the financial support from the National Natural Science Foundation of 485 China (Nos. 42002295).

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
