# Peer review of "Mapping geochemical anomalies by accounting for the uncertainty of mineralization-related elemental associations"

_EGUsphere, 2023_

## Referee Comment (RC1)

**Reviewer**

Dr Satyabrata Behera

PhD (IIT Kharagpur)

Assistant Professor, Department of Geology

Ravenshaw University, Cuttack, Odisha, India, 753003

Email Id: satyabratabehera@ravenshawuniversity.ac.in

**General Comments**

The manuscript represents a significant advancement in understanding uncertainty associated with geochemical mapping and proposes a novel methodology for the effective delineation of geochemical anomalies. It is exceptionally well-written, thoroughly researched, and offers valuable insights that will undoubtedly stimulate further research in geochemical exploration. The overall presentation is clear, concise and well-structured. With very minor revisions to address the points raised below, the manuscript has the potential to make a substantial impact and deserves consideration for publication in this prestigious international journal.

**Summary**

In this study, the concentration data of 15 elements in stream sediments were first interpolated using the inverse distance weighting (IDW) method. Then, an unsupervised soft clustering technique called fuzzy c-means (FCM) algorithm was applied to the interpolated maps. This resulted in the division of the grid cells into two clusters. Principal component analysis was applied separately to the data from the two clusters. From the resulting biplots of the first two principal components, the elemental association was determined for Cluster 1 and Cluster 2. The local singularity analysis (LSA) was performed on the IDW interpolated maps of nine elements that show strong correlations with Au in both clusters. To delineate the comprehensive anomaly, PCA was applied separately to the singularity maps of associated elements in Cluster 1 and Cluster 2. The resulting first principal component scores of both clusters that represent multi-element anomaly maps were further integrated using fuzzy membership values as weights to generate a comprehensive anomaly map which accounts for the uncertainty of

elemental association. This resulting map was then compared with the multi-element geochemical map of Au-As-Sb-Cu referred to as the "global reference case" whose element association was obtained by Zuo et al. (2021b) using Robust PCA without considering uncertainty. These two cases were compared using success rate curves and ROC-AUC plots. Further, the weights of evidence method was used to derive the statistical t-values for the multi-element anomaly scores derived from the case considering the uncertainty of elemental association and using the thresholds weak, moderate and strong anomaly regions were identified for guiding subsequent gold exploration exercise in the study area.

**Specific Comments**

**1. Page 4, Section 2 Methods:** If the "Methods" part is placed after the "Study area" part just before the Results and Discussions section, I think there will be a better flow. The authors may consider reorganising this.

**2. Page 9, Figure 3:** Regarding the Cluster Validity Index, although Xie and Beni (1991) have been cited in the text, I suggest adding a few more sentences (2-3) to the text briefly explaining the significance of this Index and how it is calculated.

**3. Page 11, Section 4.2. Mapping single- and multi-element anomalies patterns, Line 4:**

Authors may please consider citing the following article on singularity mapping.

*Behera, S., & Panigrahi, M. K. (2021). Mineral prospectivity modelling using singularity mapping and multifractal analysis of stream sediment geochemical data from the auriferous Hutti-Maski schist belt, S. India. Ore Geology Reviews, 131, 104029.*

**4. Page 14, Figure 8 (a):** It seems the "global reference map" has been created using the PC1 scores of Au-As-Sb-Cu singularity maps. However, no mention is found in the text regarding how Figure 8 (a) was generated. I suggest adding 1-2 sentences in the text regarding how the multi-element anomaly map was obtained for the global reference case.

**5. Page 14, Figure 8 (c):**

More clarification or a simplified explanation is required in the text on how the cut-off values corresponding to points A and B (Fig. 8b) were used for cases with uncertainty and global reference to generate Fig. 8c. Also, what the grey colours shown in Fig. 8 (c) indicate is not mentioned. I suggest simplifying this part for more clarity and ease of understanding.

**6. Page 15, Figure 9 (b):** Is it a box plot showing medians at the depressions? Please mention it in the text. Further, when only the top four relevant elements were retained for the two clusters, that particular case exhibited superior performance compared to the other two cases. So, from this can it be said that the elemental association Au-W-As-Sb-Ba-Hg for Cluster 1 and Au-As-W-Sn-Sb-Hg-Pb-Bi for Cluster 2 are in the decreasing order of correlation. If it is so and evident in Fig. 5 from the angles between the vectors, then it should be mentioned in the text (Page 10).

These review comments aim to acknowledge the excellence of the manuscript while providing constructive feedback to enhance its clarity and overall impact.

Thank You
Dr. Behera

---

## Author Response (AR1)

Dear Editor,

We would like to express our sincere gratitude for the opportunity to revise our manuscript. We have meticulously addressed each comment provided, ensuring that our revisions meet the esteemed standards of *Solid Earth*. Furthermore, we have conducted a thorough review of the entire manuscript, refining the language, enhancing the figures, etc., all of which help make it to align with the journal's high-quality expectations. We believe we have adequately satisfied the reviewers' insights and hope the manuscript will meet your approval for publication.

Best regards,

Jian Wang, Renguang Zuo, Qinghai Liu

**Response to Referee #1**

**General Comments**

The manuscript represents a significant advancement in understanding uncertainty associated with geochemical mapping and proposes a novel methodology for the effective delineation of geochemical anomalies. It is exceptionally well-written, thoroughly researched, and offers valuable insights that will undoubtedly stimulate further research in geochemical exploration. The overall presentation is clear, concise and well-structured. With very minor revisions to address the points raised below, the manuscript has the potential to make a substantial impact and deserves consideration for publication in this prestigious international journal.

**Summary**

In this study, the concentration data of 15 elements in stream sediments were first interpolated using the inverse distance weighting (IDW) method. Then, an unsupervised soft clustering technique called fuzzy c-means (FCM) algorithm was applied to the interpolated maps. This resulted in the division of the grid cells into two clusters. Principal component analysis was applied separately to the data from the two clusters. From the resulting biplots of the first two principal components, the elemental association was determined for Cluster 1 and Cluster 2. The local singularity analysis (LSA) was performed on the IDW interpolated maps of nine elements that show strong correlations with Au in both clusters. To delineate the comprehensive anomaly, PCA was applied separately to the singularity maps of associated elements in Cluster 1 and Cluster 2. The resulting first principal component scores of both clusters that represent multi-element anomaly maps were further integrated using fuzzy membership values as weights to generate a comprehensive anomaly map which accounts for the uncertainty of elemental association. This resulting map was then compared with the multi-element geochemical map of Au-As-Sb-Cu referred to as the "global reference case" whose element association was obtained by Zuo et al. (2021b) using Robust PCA without considering uncertainty. These two cases were compared using success rate curves and ROC-AUC plots.

Further, the weights of evidence method was used to derive the statistical t-values for the multielement anomaly scores derived from the case considering the uncertainty of elemental association and using the thresholds weak, moderate and strong anomaly regions were identified for guiding subsequent gold exploration exercise in the study area.

**Reply**: We are immensely grateful for your encouraging and insightful comments regarding our manuscript. Your summary captures the essence of our work and further reinforces its potential contribution to the field of geochemical mapping. We are committed to addressing the revisions you have suggested and hope that the revisions made will render the manuscript suitable for publication in *Solid Earth*.

Please find below our point-by-point responses to your specific comments:

(1) Page 4, Section 2 Methods: If the "Methods" part is placed after the "Study area" part just before the Results and Discussions section, I think there will be a better flow. The authors may consider reorganising this.

**Reply**: Thank you for your suggestion. We have carefully considered the structure of our manuscript and agree that placing the "Methods" part after the "Study area" part could indeed enhance the narrative flow and provide a more logical progression into the "Results and Discussions" section. We will implement this change to improve the coherence and readability of our manuscript.

**Changes in manuscript**: We have changed the order of the "Methods" part and the "Study area and data" part. Please refer to the revised manuscript for more information.

(2) Page 9, Figure 3: Regarding the Cluster Validity Index, although Xie and Beni (1991) have been cited in the text, I suggest adding a few more sentences (2-3) to the text briefly explaining the significance of this Index and how it is calculated.

**Reply**: Thank you for your constructive feedback. We have revised the manuscript to include a brief explanation of the Cluster Validity Index in section 3.2 ("Fuzzy c-means clustering"), highlighting its role in optimizing cluster number by assessing compactness and separation, as proposed in Xie and Beni (1991). Specifically, the index gauges the quality of clustering; a lower value indicates denser, more distinct clusters.

**Changes in manuscript**: We added the following part in section 3.2 in the revised manuscript: "*In this study, we employed the Xie-Beni Validity Index (Xie and Beni, 1991) to determine the optimal cluster number, which is defined as*

$$S = \frac{\sum_{i=1}^{C} \sum_{j=1}^{N} \mu_{ij}^2 D_{ij}}{N \left( \min_{i,j=1,\cdots C, i \neq j} \|c_i - c_j\|^2 \right)}$$

*This index evaluates the dataset's geometric structure and membership degrees, offering a measure of cluster compactness and separation. A lower index value signifies elevated cluster density and distinction*".

(3) Page 11, Section 4.2. Mapping single- and multi-element anomalies patterns, Line 4: Authors may please consider citing the following article on singularity mapping.

Behera, S., & Panigrahi, M. K. (2021). Mineral prospectivity modelling using singularity mapping and multifractal analysis of stream sediment geochemical data from the auriferous Hutti-Maski schist belt, S. India. Ore Geology Reviews, 131, 104029.

**Reply**: Upon thorough examination, we find this resource to be pertinent to our study. This addition undoubtedly enhances our manuscript by providing a broader context and supporting our methodology effectively.

**Changes in manuscript**: We have incorporated this reference into the revised manuscript, acknowledging its contribution to the foundational theory and application of singularity mapping in geochemical mapping. Please see line 297 in the revised manuscript for more information.

(4) Page 14, Figure 8 (a): It seems the "global reference map" has been created using the PC1 scores of Au-As-Sb-Cu singularity maps. However, no mention is found in the text regarding how Figure 8 (a) was generated. I suggest adding 1-2 sentences in the text regarding how the multi-element anomaly map was obtained for the global reference case.

**Reply**: Thank you for your suggestion. We have now included a description in the manuscript, see the first paragraph in section 4.3, to clarify the derivation of the multi-element anomaly map. The global reference map was constructed using the first principal component scores from the singularity maps of Au, As, Sb, and Cu. This method mirrors the approach used for creating multivariate anomaly maps for each elemental association for the case with uncertainty considered, ensuring consistency across our analyses.

**Changes in manuscript**: The improved description in section 4.3 (lines 341-345 in the revised manuscript) is: "*The multi-element anomaly patterns were derived consistently with those for each elemental association in the case with uncertainty considered. This was achieved by applying PCA to the univariate anomaly maps of Au, As, Sb, and Cu, and retaining the first principal component to represent the multivariate anomaly map (Fig. 8a). It exhibits strong spatial correlation with the geological structures and known gold mineral deposits.*"

(5) Page 14, Figure 8 (c): More clarification or a simplified explanation is required in the text on how the cut-off values corresponding to points A and B (Fig. 8b) were used for cases with uncertainty and global reference to generate Fig. 8c. Also, what the grey colours shown in Fig. 8 (c) indicate is not mentioned. I suggest simplifying this part for more clarity and ease of understanding.

**Reply**: Thank you for your feedback. Figure 8(c) was used to visually discern the disparities in the spatial distribution of geochemical patterns that contribute to the different performance observed between the two points A and B. Considering that a specific anomaly area can be defined for a given threshold, such geochemical pattern corresponds to the incremental areas delimited by the two cut-off values corresponding to points A and B, respectively. We have revised the text to provide a clearer and more straightforward explanation of how the cut-off values from points A and B in Figure 8 (b) were applied to the cases with

uncertainty and the global reference to produce Figure 8c. Additionally, we have updated Figure 8 to enhance its informativeness, and removed the grey hillshade background in Figure 8c to improve clarity. We believe these revisions have made the figure's interpretation more intuitive and the methodology more transparent.

**Changes in manuscript**: The improved description in section 4.3 (lines 359-366 in the revised manuscript) is: "*By examining the incremental areas delineated by the cut-off values corresponding to points A and B for the two scenarios, we were able to visually discern the disparities in the spatial distribution of geochemical patterns that contribute to the different performance observed between these two points. This is achieved by subtracting the cumulative area corresponding to the threshold defined by point A from that of point B for each scenario, thus isolating the specific regions responsible for the discrepancy in performance (Fig. 8c). The global reference case placed a greater emphasis on the southeastern region, as indicated by the dashed rectangle in Fig. 8c, which is distinguished by elevated geochemical backgrounds for most indicative elements due to the prevalence of Paleozoic gold-enrichment lithologic units*".

(6) Page 15, Figure 9 (b): Is it a box plot showing medians at the depressions? Please mention it in the text. Further, when only the top four relevant elements were retained for the two clusters, that particular case exhibited superior performance compared to the other two cases. So, from this can it be said that the elemental association Au-W-As-Sb-Ba-Hg for Cluster 1 and Au-As-W-Sn-Sb-Hg-Pb-Bi for Cluster 2 are in the decreasing order of correlation. If it is so and evident in Fig. 5 from the angles between the vectors, then it should be mentioned in the text (Page 10).

**Reply**: Thank you for your insightful comments. We confirm that the box plot indeed shows medians at the notches, and we have now explicitly mentioned this in the revised manuscript. The notches serve as a visual representation of the median's confidence interval, and the lack of overlap between the notches for the two cases underscores a statistically significant difference in their performance. This addition strengthens the evidence for the superiority of the case accounting for uncertainty.

In addition, we have taken your constructive feedback into account and have clarified in the text that the elemental associations for Cluster 1 and Cluster 2 are presented in descending order of correlation with the ore-forming element Au. This revision is now reflected in the last paragraph of section 4.1.

**Changes in manuscript**: (1) We add the following sentence to the caption of Figure 9: "*Note that the notched boxplot applies a "notch" around the median, which serves as a visual representation of the median's confidence interval.*"; (2) we add the following sentence to the text in section 4.4 (lines 393-394): "*The non-overlapping notches of the boxes signify a statistically significant median difference between the two cases.*"; (3) we revised the sentence (lines 283-285) about defining elementals associations as "*For Cluster 1, the elemental association identified, in descending order of correlation with the ore-forming element Au, is Au-W-As-Sb-Ba-Hg. Similarly, for Cluster 2, the sequence is Au-As-W-Sn-Sb-Hg-Pb-Bi, as evidenced by the biplots in Figs. 5a and b.*".

**Response to Referee #2**

This is an interesting study offering a novel tool for dealing with uncertainty in geochemical mapping. The manuscript could be published with minor revisions.

**Reply**: Thank you for your constructive and encouraging feedback on our study. We are grateful for your recognition of the study's novelty and its contribution to geochemical mapping. We have addressed your comments that help improve the quality and clarity of our work.

Please find below our point-by-point responses to your specific comments:

(1) "The introduction is well-written, providing the reader with the state-of-the-art of geochemical mapping. Yet, the second page was missing, meaning that my review is exclusive of the 2$^{nd}$."

**Reply**: Thank you for your insightful comments on our manuscript's introduction. Regarding the missing second page, we have reexamined the preprint version and confirmed that it is complete. We suspect there may have been a technical glitch during the download or viewing process. We apologize for any inconvenience this may have caused.

(2) "The authors try to "reduce the uncertainty" in geochemical mapping, yet there is no mention of the type of uncertainty they try to deal with in the introduction."

**Reply**: Thank you for your valuable comment. Our manuscript specifically addresses the uncertainty arising from determining elemental associations indicative of geological process of interest, such as mineralization. In the section of introduction, we demonstrate the utility of elemental associations in representing geological processes of interest, particularly in the third to last paragraph. Furthermore, we discuss the inherent uncertainty of this practice due to the spatial and temporal heterogeneity of geological processes in the second to last paragraph of the introduction.

**Changes in manuscript**: To enhance the manuscript's clarity, we have revised the expressions in the last paragraph and explicitly underscored the type of uncertainty our study aims to reduce more emphatically, namely (lines 98-100): "*To mitigate the uncertainty inherent in defining elemental associations, this study introduces a workflow that utilizes fuzzy clustering to delineate homogeneous zones and further determines their respective elemental associations, complemented by PCA and geochemical anomaly detection techniques for refined geochemical mapping*".

(3) "Section 2.1: It is not clear as to how the proposed workflow deals with 'uncertainty'."

**Reply**: Thank you for your comment. In the revised manuscript, we have taken steps to further clarify how our proposed workflow addresses the uncertainty arising from defining elemental associations in geochemical mapping.

**Changes in manuscript**:

- We have added explanatory sentences at the beginning of Section 2.1, outlining the general idea our workflow takes to tackle the uncertainty associated with defining elemental associations. (The added sentences are: "*To effectively address the uncertainty in defining elemental associations for geochemical mapping, our workflow starts by employing clustering analysis to pinpoint homogeneous regions, each presumed to be characterized by a distinct elemental association. Subsequently, it ascertains the pertinent elemental associations for each identified cluster, with the ensemble of these associations representing the uncertainty. The workflow proceeds by performing multivariate geochemical anomaly mapping for each potential elemental association, ultimately synthesizing a comprehensive geochemical map through a linear weighting scheme based on the alternative maps.*")
- We have revised the descriptions of the relevant steps in our workflow to more explicitly articulate how it addresses the uncertainty.
- We have also revised Figure 1 in original manuscript to better highlight both the type of uncertainty we aim to address and the core steps our workflow employs to deal with it.

[Figure]

We believe these improvements will make the methodology and its relation to uncertainty more transparent to the reader.

(4) "Section 2.3: The ''Hadamard product'' requires more clarity. What is the range of membership value that feeds equation 3? Have the authors applied scaling to their Ai maps?"

**Reply**: Thank you for your suggestion for further clarification on the Hadamard product and its application in our study. Please find below our responses to your comments:

- Regarding the "Hadamard product", we have further elaborated on it in Section 2.3, emphasizing its theoretical simplicity as a binary operation applied to two matrices of identical dimensions, which

results in a matrix where each element is the product of the corresponding elements from the original matrices.

- The range of membership values used in the Hadamard product has been explicitly detailed in Section 2.2 in the revised manuscript. We would like to emphasize that the usage of membership values, which range from 0 to 1, in merging multiple anomaly maps hinges upon the key understanding that every membership value signifies a corresponding degree of belonging of each cell sample to the clusters. This also constitutes an essential aspect of using fuzzy clustering for addressing uncertainty stemming from defining elemental associations.

- In relation to the question on scaling of our "Ai maps", we have not performed any such scaling on the anomaly scores considering that the singularity exponent by itself is a measure without any dimensions. However, we do recognize that our methodology could well be applied to anomaly scores that are not dimensionless if they're used in conjunction with normalization techniques.

**Changes in manuscript**: The improved description for "Hadamard product" is (lines 218-225): *When considering two matrices, designated as A and B, both of which have identical dimensions m by n, one can compute the Hadamard product $A \odot B$. This results in a matrix with matching dimensions, where each element $(A \odot B)_{ij}$ is the product of the corresponding elements from A and B, namely $(A \odot B)_{ij} = A_{ij} \times B_{ij}$. Note that the membership values, ranging from 0 to 1, serve as a quantitative representation of each cell sample's affiliation with the identified clusters (hence the corresponding elemental associations), thereby facilitating the integration of multiple anomaly maps through the Hadamard product. Moreover, while our anomaly scores, derived from the singularity exponent, require no scaling due to their dimensionless nature, we acknowledge that the Hadamard product can also be applied to scaled anomaly scores, provided they are normalized to ensure comparability across different measures.*